# The Edge-of-Reach Problem in Offline Model-Based Reinforcement Learning

**Anya Sims**
University of Oxford
anya.sims@stats.ox.ac.uk

**Cong Lu**
University of Oxford

**Jakob N. Foerster**
FLAIR, University of Oxford

**Yee Whye Teh**
University of Oxford

## Abstract

Offline reinforcement learning (RL) aims to train agents from pre-collected datasets. However, this comes with the added challenge of estimating the value of behaviors not covered in the dataset. Model-based methods offer a potential solution by training an approximate dynamics model, which then allows collection of additional synthetic data via rollouts in this model. The prevailing theory treats this approach as online RL in an approximate dynamics model, and any remaining performance gap is therefore understood as being due to dynamics model errors. In this paper, we analyze this assumption and investigate how popular algorithms perform as the learned dynamics model is improved. In contrast to both intuition and theory, *if the learned dynamics model is replaced by the true error-free dynamics, existing model-based methods completely fail.* This reveals a key oversight: The theoretical foundations assume sampling of full horizon rollouts in the learned dynamics model; however, in practice, the number of model-rollout steps is aggressively reduced to prevent accumulating errors. We show that this truncation of rollouts results in a set of edge-of-reach states at which we are effectively "bootstrapping from the void." This triggers pathological value overestimation and complete performance collapse. We term this the edge-of-reach problem. Based on this new insight, we fill important gaps in existing theory, and reveal how prior model-based methods are primarily addressing the edge-of-reach problem, rather than model-inaccuracy as claimed. Finally, we propose *Reach-Aware Value Learning* (RAVL), a simple and robust method that directly addresses the edge-of-reach problem and hence - unlike existing methods - does not fail as the dynamics model is improved. Since world models will inevitably improve, we believe this is a key step towards future-proofing offline RL.[1]

## 1   Introduction

Standard online reinforcement learning (RL) requires collecting large amounts of on-policy data. This can be both costly and unsafe, and hence represents a significant barrier against applying RL in domains such as healthcare [28, 32] or robotics [2, 4, 20], and also against scaling RL to more complex problems. Offline RL [6, 23] aims to remove this need for online data collection by enabling agents to be trained on pre-collected datasets. One hope [21] is that it may facilitate advances in RL similar to those driven by the use of large pre-existing datasets in supervised learning [3].

---

[1]Our code is open-sourced at: github.com/anyasims/edge-of-reach.

38th Conference on Neural Information Processing Systems (NeurIPS 2024).

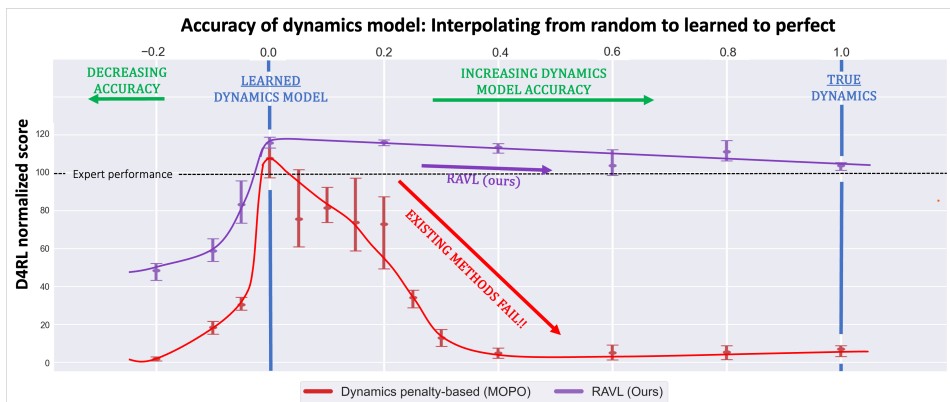

Figure 1: *Existing offline model-based RL methods fail if the accuracy of the dynamics model is increased (with all else kept the same)*. Results shown are for MOPO [36], but note that this failure indicates the *failure of all existing uncertainty-based methods* since each of their specific penalty terms disappear under the true dynamics as 'uncertainty' is zero. By contrast, our method is much more robust to changes in dynamics model. The $x$-axis shows linearly interpolating next states and rewards of the learned model with the true model (center→right) and random model (center→left), with results on the D4RL W2d-medexp benchmark (min/max over 4 seeds). The full set of results and experimental setup are provided in Table 1 and Appendix C.2 respectively.

The central challenge in offline RL is estimating the value of actions not present in the dataset, known as the *out-of-sample action* problem [17].[2] A naïve approach results in extreme value overestimation due to bootstrapping using inaccurate values at out-of-sample state-actions [18]. There have been many proposals to resolve this, with methods largely falling into one of two categories: model-free [1, 8, 10, 17–19] or model-based [14, 24, 29, 36].

Model-free methods typically address the out-of-sample action problem by applying a form of conservatism or constraint to avoid using out-of-sample actions in the Bellman update. In contrast, the solution proposed by model-based methods is to allow the collection of additional data at any previously out-of-sample actions. This is done by first training an approximate dynamics model on the offline dataset [13, 30], and then allowing the agent to collect additional synthetic data in this model via $k$-step rollouts (see Algorithm 1). The prevailing understanding is that this can be viewed as online RL in an approximate dynamics model, with the instruction being to then simply "run any RL algorithm on $\widehat{M}$ until convergence"[36], where $\widehat{M}$ is the learned dynamics model with some form of dynamics uncertainty penalty. Existing methods propose various forms of dynamics penalties (see Table 7), based on the assumption that the remaining performance gap compared to online RL is solely due to inaccuracies in the learned dynamics model.

This understanding naturally implies that improving the dynamics model should also improve performance. Surprisingly, we find that existing offline model-based methods completely fail if the learned dynamics model is replaced with the true, error-free dynamics model, while keeping everything else the same (see Figure 1). Under the true dynamics, the only difference to online RL is that in online RL, data is sampled as full-length episodes, while in offline model-based RL, data is instead sampled as $k$-step rollouts, starting from a state in the original offline dataset, with rollout length $k$ limited to avoid accumulating model errors. Failure under the true model therefore highlights that truncating rollouts has critical and previously-overlooked consequences.

We find that this rollout truncation leads to a set of states which, under any policy, can only be reached in the final rollout step (see red in Figure 2). The existence of these *edge-of-reach* states is problematic as it means Bellman updates (see Equation (1)) use target values that are never themselves trained. This is illustrated in Figure 2, and described in detail in Section 3. This effective "bootstrapping from the void" triggers a catastrophic breakdown in $Q$-learning. Concisely, this issue can be viewed as all actions from edge-of-reach states remaining out-of-reach over training. Hence, contrary to common understanding, the out-of-sample action problem central to model-free methods is not fully resolved by a model-based approach. In fact, in Section 3.4 we provide detailed analysis suggesting that this is the predominant source of issues on the standard D4RL benchmark. In Section 6.5 we consequently reexamine how existing methods work and find they are indirectly and unintentionally addressing this issue (rather than model errors as claimed).

---

[2]This is also referred to as the *out-of-distribution action* [18] or *action distribution shift* problem [19].

Algorithm 1: Pseudocode for the base procedure used in offline model-based methods.
*In summary: (1) Train a dynamics model $\widehat{M}$ on $\mathcal{D}_{\text{offline}}$, then (2) Train an agent (often SAC) with $k$-step rollouts in the learned model starting from $s \in \mathcal{D}_{\text{offline}}$.* Existing methods consider issues to be due to errors in $\widehat{M}$ and hence introduce **dynamics uncertainty penalties** (see Table 7). We find the predominant source of issues to be the edge-of-reach problem, and hence instead propose using **value pessimism (RAVL)** (see Section 5).

---

**Algorithm 1** Base model-based algorithm (MBPO) + Additions in **existing methods** and **RAVL (ours)**

---

1: **Require:** Offline dataset $\mathcal{D}_{\text{offline}}$
2: **Require:** Dynamics model $\widehat{M} = (\widehat{T}, \widehat{R})$ trained on $\mathcal{D}_{\text{offline}}$. **Augment $\widehat{M}$ with an uncertainty penalty***
3: **Specify:** Rollout length $k \geq 1$, real data ratio $r \in [0, 1]$
4: **Initialize:** Replay buffer $\mathcal{D}_{\text{rollouts}} = \emptyset$, policy $\pi_\theta$, value function $Q_\phi$ (both from random)
5: **for** epochs $= 1, \ldots$ **do**
6:     **(Collect data)** Starting from states in $\mathcal{D}_{\text{offline}}$, collect $k$-step rollouts in $\widehat{M}$ with $\pi_\theta$. Store data in $\mathcal{D}_{\text{rollouts}}$
7:     **(Train agent)** Train $\pi_\theta$ and $Q_\phi$ on $\mathcal{D}_{\text{rollouts}} \cup \mathcal{D}_{\text{offline}}$ (mixed with ratio $r$) **Add $Q$-value pessimism**
8: **end for**    *This uncertainty penalty collapses to zero with the true error-free dynamics ($\widehat{M} = M$) (see experiments Table 2).*

---

*We have the following problem:* Indirectly addressing the edge-of-reach problem means existing model-based methods have a fragile and unforeseen dependence on model quality and fail catastrophically as dynamics models improve. Since model improvements are inevitable, and since tuning of hyperparameters is particularly impractical in offline RL, this is a substantial practical barrier.

In light of this, we propose *Reach-Aware Value Learning* (RAVL), a simple and robust method that directly addresses the edge-of-reach problem. RAVL achieves strong performance on standard benchmarks, and moreover continues to perform even with error-free, uncertainty-free dynamics models (Section 6). Our model-based method bears close connections to model-free approaches, and hence in Appendix A.2 we present a unified perspective of these two previously disjoint subfields.

## 2 Background

**Reinforcement Learning.** We consider the standard RL framework [31], in which the environment is formulated as a Markov Decision Process, $M = (\mathcal{S}, \mathcal{A}, T, R, \mu_0, \gamma)$, where $\mathcal{S}$ and $\mathcal{A}$ denote the state and action spaces, $T(s'|s, a)$ and $R(s, a)$ denote the transition and reward dynamics, $\mu_0$ the initial state distribution, and $\gamma \in (0, 1)$ is the discount factor. The goal in reinforcement learning is to learn a policy $\pi(a|s)$ that maximizes the expected discounted return $\mathbb{E}_{\mu_0, \pi, T}\left[\sum_{t=0}^{\infty} \gamma^t R(s_t, a_t)\right]$.

**Actor-Critic Algorithms.** The broad class of algorithms we consider are actor-critic [15] methods which jointly optimize a policy $\pi$ and state-action value function ($Q$-function). Given a dataset $\mathcal{D}$ of (*state, action, reward, nextstate*) transitions, actor-critic algorithms iterate between two steps:

---

1. *(Policy evaluation)* Update $Q_\phi$ to fit the current policy using the following Bellman update:

   **Key expression:**     $\underbrace{Q_\phi(s, a)}_{\text{Update value at } (s, a)} \leftarrow \underbrace{r + \gamma Q_\phi(s', \pi_\theta(s'))}_{\text{Using value at } (s', \pi_\theta(s'))}$     $(s, a, r, s') \sim \mathcal{D}$     (1)

---

2. *(Policy improvement)* Update $\pi_\theta$ to increase the current $Q$-value predictions: $Q_\phi(s, \pi_\theta(s))$[3].

**Offline RL and the Out-of-Sample Action Problem.** In *offline* RL, training must rely on only a fixed dataset of transitions $\mathcal{D}_{\text{offline}} = \{(s, a, r, s')^{(i)}\}_{i=1,\ldots,N}$ collected by some policy $\pi^\beta$. The central problem in this offline setting is the out-of-sample[4] action problem: The values of $Q_\phi$ being updated are at any state-actions that appear as $(s, a)$ in $\mathcal{D}_{\text{offline}}$. However, the targets of these updates rely on values at $(s', a')$ where $a' \sim \pi_\theta$ (see Equation (1)). Since $a' \sim \pi_\theta$ (on-policy) whereas $a \sim \pi^\beta$ (off-policy), $(s', a')$ may not appear in $\mathcal{D}_{\text{offline}}$, and hence may itself have never been updated. This means that updates can involve bootstrapping from 'out-of-sample' and hence arbitrarily misestimated values and thus lead to misestimation being propagated over the entire state-action space. The max operation (implicit in the policy improvement step) further acts to exploit any misestimation, converting misestimation into extreme pathological overestimation. As a result $Q$-values often tend to increase unboundedly over training, while performance collapses entirely [18, 23].

---

[3]This step contains an implicit max operation which we refer to in Section 3.2.
[4]We use the terms 'out-of-sample' and 'out-of-distribution' interchangeably, as is done in the literature.

**Model-Based Offline RL.** Model-based methods [30] aim to solve the out-of-sample issue by allowing the agent to collect additional synthetic data in a learned dynamics model. They generally share the same base procedure as described in Algorithm 1. This involves first training an approximate dynamics model $\widehat{M} = (\mathcal{S}, \mathcal{A}, \widehat{T}, \widehat{R}, \mu_0, \gamma)$ on $\mathcal{D}_{\text{offline}}$. Here, $\widehat{T}(s'|s, a)$ and $\widehat{R}(s, a)$ denote the learned transition and reward functions, commonly realized as a deep ensemble [5, 22]. Following this an agent is trained using an online RL algorithm (typically SAC [11]), for which data is sampled as $k$-step trajectories (termed rollouts) under the current policy, starting from states in the offline dataset $\mathcal{D}_{\text{offline}}$. The base procedure of training a SAC agent with model rollouts does not work out of the box. Existing methods attribute this as due to dynamics model errors. Consequently, a broad class of methods propose augmenting the learned dynamics model $\widehat{M}$ with some form of dynamics uncertainty penalty, often based on variance over the ensemble dynamics model [2, 14, 24, 29, 36]. We present some explicit examples in Table 7. Crucially, all of these methods assume that, under the true error-free model, no intervention should be needed, and hence all the uncertainty penalties collapse to zero. In the following section, we show that this assumption leads to catastrophic failure.

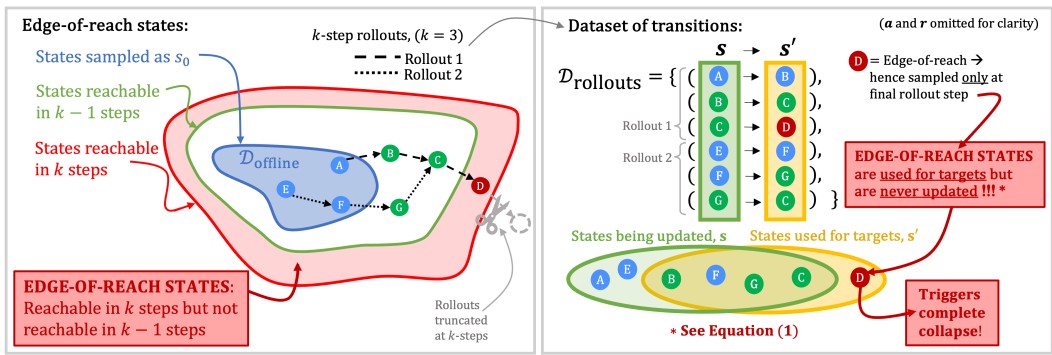

Figure 2: The previously unnoticed edge-of-reach problem. *Left* illustrates the base procedure used in offline model-based RL, whereby synthetic data is sampled as $k$-step trajectories "rollouts" starting from a state in the original offline dataset. Edge-of-reach states are those that can be reached in $k$-steps, but which cannot (under any policy) be reached in less than $k$-steps. We depict the data collected with two rollouts, one ending in $s_k = D$, and the other with $s_k = C$. *Right* then shows this data arranged into a dataset of transitions as used in $Q$-updates. State $D$ is edge-of-reach and hence appears in the dataset as $s'$ *but never as $s$.* Bellman updates therefore *bootstrap from $D$, but never update the value at $D$* (see Equation (1)). (For comparison consider state $C$: $C$ is also sampled at $s_k$, but unlike $D$ it is not edge-of-reach, and hence is also sampled at $s_{i<k}$ meaning it *is* updated and hence does not cause issues.)

## 3 The Edge-of-Reach Problem

In the following section, we formally introduce the *edge-of-reach* problem. We begin by showing the empirical failure of SOTA offline model-based methods on the *true environment dynamics*. Next, we present our edge-of-reach hypothesis for this, including intuition, empirical evidence on the main D4RL benchmark [7], and theoretical proof of its effect on offline model-based training.

### 3.1 Surprising Failure with the True Dynamics

As described in Sections 1 and 2, prior works view offline model-based RL as *online RL in an approximate dynamics model*. Based on this understanding they propose various forms of dynamics uncertainty penalties to address model errors (see Table 7). This approach is described simply as: *"two steps: (a) learning a pessimistic MDP (P-MDP) using the offline dataset; (b) learning a near-optimal policy in this P-MDP"*[14]. This shared base procedure is shown in Algorithm 1, where *"P-MDP"* refers to the learned dynamics model with a penalty added to address model errors.

This assumption of issues being due to dynamics model errors naturally leads to the belief that the ideal case would be to have a perfect error-free dynamics model. However, in Figure 1 and Table 1 we demonstrate that, *if the learned dynamics model is replaced with the true dynamics,*

*all dynamics-penalized methods completely fail on most environments.*[5] Note that under the true dynamics, all existing dynamics penalty-based methods assume no intervention is needed since there are no model errors. As a result, their penalties all become zero (see Table 7) and they all collapse to exactly the same procedure. Therefore the results shown in Table 1 indicate the failure of *all* existing dynamics-penalized methods. In the following section, we investigate why having perfectly accurate dynamics leads to failure.

## 3.2 The Edge-Of-Reach Hypothesis (Illustrated in Figure 2)

Failure under the error-free dynamics reveals that "model errors" cannot explain all the problems in offline model-based RL. Instead, it highlights that there must be a second issue. On investigation we find this to be the "edge-of-reach problem." We begin with the main intuition: in offline model-based RL, synthetic data $\mathcal{D}_{\text{rollouts}}$ is generated as short $k$-step rollouts starting from states in the original dataset $\mathcal{D}_{\text{offline}}$ (see Algorithm 1). Crucially, **Figure 2** *(left)* illustrates how, under this procedure, there can exist some states which can be reached in the final rollout step, but which *cannot* - under any policy - be reached earlier. These *edge-of-reach* states triggers a breakdown in learning, since even with the ability to collect unlimited data, the agent is never able to reach these states 'in time' to try actions from them, and hence is free to 'believe that these edge-of-reach states are great.'

More concretely: **Figure 2** *(right)* illustrates how being sampled only at the final rollout step means edge-of-reach states will appear in the resulting dataset $\mathcal{D}_{\text{rollouts}}$ as *nextstates* $s'$, but will never appear in the dataset as *states* $s$. Crucially, in Equation (1), we see that values at states $s$ are updated, while values at $s'$ are used for the targets of these updates. Edge-of-reach states are therefore *used for targets, but are never themselves updated*, meaning that their values can be arbitrarily misestimated. Updates consequently propagate misestimation over the entire state-action space. Furthermore, the max operation in the policy improvement step (see Section 2) *exploits any misestimation* by picking out the most heavily overestimated values [23]. The *misestimation* is therefore turned into *overestimation*, resulting in the value explosion seen in Figure 3. Thus, contrary to common understanding, the out-of-sample action problem key in model-free offline RL can be seen to persist in model-based RL.

> **Edge-of-Reach Hypothesis**: Limited horizon rollouts from a fixed dataset leads to 'edge-of-reach' states: states which are used as targets for Bellman-based updates, but which are never themselves updated. The resulting maximization over misestimated values in Bellman updates causes pathological value overestimation and a breakdown in $Q$-learning.

***When is this an issue in practice?*** Failure via this mode requires *(a)* the existence of edge-of-reach states, and *(b)* such edge-of-reach states to be sampled. The typical combination of $k \ll H$ along with a limited pool of starting states ($s \in \mathcal{D}_{\text{offline}}$) means the rollout distribution is unlikely to sufficiently cover the full state space $\mathcal{S}$, thus making *(a)* likely. Moreover, we observe pathological 'edge-of-reach seeking' behavior in which the agent appears to 'seek out' edge-of-reach states due to this being the source of overestimation, thus making *(b)* likely. This behaviour is discussed further in Section 4.

In Appendix A, we give a thorough and unified view of model-free and model-based offline RL, dividing the problem into independent conditions for states and actions and examining when we can expect the edge-of-reach problem to be significant.

## 3.3 Definitions and Formalization

**Definition 1** (Edge-of-reach states). *Consider a deterministic transition model $T : \mathcal{S} \times \mathcal{A} \to \mathcal{S}$, rollout length $k$, and some distribution over starting states $\nu_0$. For some policy $\pi : \mathcal{S} \to \mathcal{A}$, rollouts are then generated according to $s_0 \sim \nu_0(\cdot)$, $a_t \sim \pi(\cdot|s_t)$ and $s_{t+1} \sim T(\cdot|s_t, a_t)$ for $t = 0, \ldots, k-1$, giving $(s_0, a_0, s_1, \ldots, s_k)$. Let us use $\rho_{t,\pi}(s)$ to denote the marginal distributions over $s_t$.*

*We define a state $s \in S$ **edge-of-reach** with respect to $(T, k, \nu_0)$ if: for $t = k$, $\exists \pi$ s.t. $\rho_{t,\pi}(s) > 0$, but, for $t = 1, \ldots, k-1$ and $\forall \pi$, $\rho_{t,\pi}(s) = 0$. In our case, $\nu_0$ is the distribution of states in $\mathcal{D}_{\text{offline}}$.*

In Appendix B, we include an extension to stochastic transition models, proof of how errors can consequently propagate to all states, and a discussion of the practical implications.

---

[5]For clarity, this change (**Approximate**→**True**) is described in Algorithm 1 as pseudocode, and also in Appendix H in terms of how it relates to other experiments presented throughout the paper.

### 3.4 Empirical evidence on the D4RL benchmark

There are two potential issues in offline model-based RL: *(1)* dynamics model errors and subsequent model exploitation, and *(2)* the edge-of-reach problem and subsequent pathological value overestimation. We ask: *Which is the true source of the issues observed in practice?* While *(1)* is stated as the sole issue and hence the motivation in prior methods, the "failure" results presented in Figure 1 and Table 1 are strongly at odds with this explanation. Furthermore, in Table 8, we examine the model rewards sampled by the agent over training. For *(1)* to explain the $Q$-values seen in Figure 3, the sampled rewards in Table 8 would need to be on the order of $10^8$. Instead, however, they remain less than 10, and are not larger than the true rewards, meaning *(1)* does not explain the observed value explosion (see Appendix D.2). By contrast, the edge-of-reach-induced pathological overestimation mechanism of *(2)* exactly predicts this value explosion, with the observations very closely resembling those of the analogous model-free out-of-sample problem [18]. Furthermore, *(2)* is consistent with the "failure" observations in Table 1. Finally, we again highlight the discussion in Section 3.2 where we explain why the edge-of-reach problem can be expected to occur in practice.

Figure 3: The base procedure results in poor performance *(left)* with exponential increase in $Q$-values *(right)* on the D4RL benchmark. *Approx $Q^*$* indicates the $Q$-value for a normalized score of 100 (with $\gamma = 0.99$). Results are shown for Walker2d-medexp (6 seeds), but we note similar trends across other D4RL datasets.

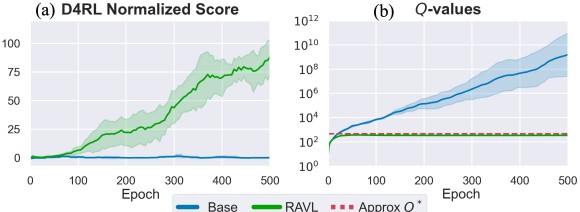

## 4 Analysis with a Simple Environment

In the previous section, we presented the edge-of-reach hypothesis as an explanation of why existing methods fail under the true dynamics. In this section, we construct a simple environment to empirically confirm this hypothesis. We first reproduce the observation seen in Figures 1 and 3 and Table 1 of the base offline model-based procedure resulting in exploding $Q$-values and failure to learn despite using the true dynamics model. Next, we verify that edge-of-reach states are the source of this problem by showing that correcting value estimates only at these states is sufficient to resolve the issues.

### 4.1 Setup

We isolate failure observed in Section 3.1 with the following setup: Reward is defined as in Figure 4a, the transitions function is simply $\mathbf{s'} = \mathbf{s} + \mathbf{a} \in \mathbb{R}^2$, and initial states (analogous to $s \in \mathcal{D}_{\text{offline}}$) are sampled from $\mu_0 = U([-2, 2]^2)$ (the area shown in the navy blue box Figure 4).

In applying the offline model-based procedure (see Algorithm 1) to this (true) environment we have precisely the same setup as in Section 3.1, where again the *only difference compared to online RL is the use of truncated $k$-step rollouts* with $k = 10$ (compared to full horizon $H = 30$). This small change results in the existence of edge-of-reach states (those between the red and orange boxes).

### 4.2 Observing Pathological Value Overestimation

Exactly as with the benchmark experiments in Section 3.1, the base model-based procedure fails despite using the true dynamics (see **blue** Figure 4, **Base**), and the $Q$-values grow unboundedly over training (compare Figure 4f and Figure 3b).

Looking at the rollouts sampled over training (see Figure 7) we see the following behavior:

*(Before 25 epochs)* Performance initially increases. *(Between 25 and 160 epochs)* Value misestimation takes over, and the policy begins to aim toward unobserved state-actions (since their values can be misestimated and hence overestimated). *(After 160 epochs)* This *'edge-of-reach seeking' behavior* compounds with each epoch, leading the agent to eventually reach edge-of-reach states. From this point onwards, the agent samples edge-of-reach states at which it never receives any corrective feedback. The consequent pathological value overestimation results in a complete collapse in performance. In Figure 4b we visualize the final policy and see that it completely ignores the reward function, aiming instead towards an arbitrary edge-of-reach state.

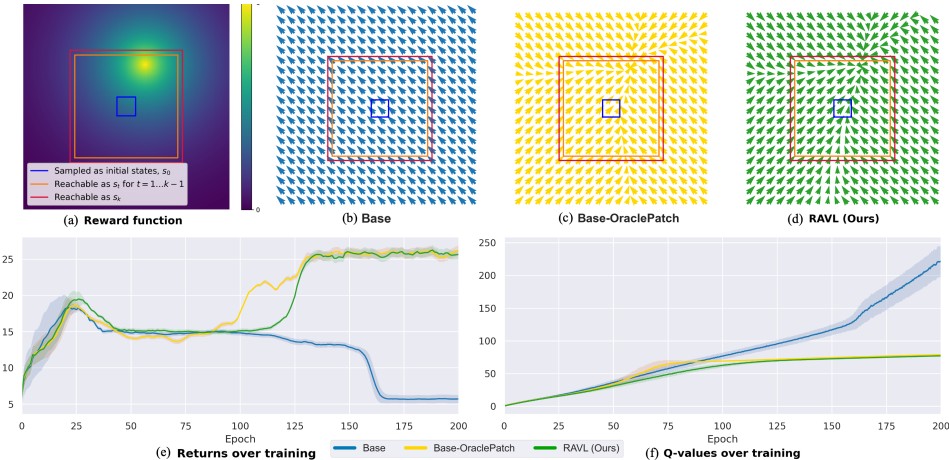

Figure 4: Experiments on the simple environment, illustrating the edge-of-reach problem and potential solutions. **(a)** Reward function, **(b)** final (failed) policy with naïve application of the base procedure (see Algorithm 1), **(c)** final (successful) policy with patching in oracle $Q$-values for edge-of-reach states, **(d)** final (successful) policy with RAVL, **(e)** returns evaluated over training, **(f)** mean $Q$-values evaluated over training.

## 4.3 Verifying the Hypothesis Using Value Patching

Our hypothesis is that the source of this failure is value misestimation at edge-of-reach states. Our **Base-OraclePatch** experiments (see **yellow** Figure 4) verify this by showing that patching in the correct values solely at edge-of-reach states is sufficient to completely solve the problem. This is particularly compelling as in practice we only corrected values at 0.4% of states over training. In Section 5 we introduce our practical method RAVL, which Figures 4 **green** and 7 show has an extremely similar effect to that of the ideal but practically impossible Base-OraclePatch intervention.

## 5 RAVL: Reach-Aware Value Learning

As verified in Section 4.3, issues stem from value overestimation at edge-of-reach states. To resolve this, we therefore need to *(A)* detect and *(B)* prevent overestimation at edge-of-reach states.

*(A) Detecting edge-of-reach states:* As illustrated in Figure 2 *(right)*, edge-of-reach states are states at which the $Q$-values are never updated, i.e., those that are out-of-distribution (OOD) with respect to the training distribution of the $Q$-function, $s \in \mathcal{D}_{\text{rollouts}}$. A natural solution for OOD detection is measuring high variance over a deep ensemble [22]. We can therefore detect edge-of-reach states using an ensemble of $Q$-functions. We demonstrate that this is effective in Figure 5.

*(B) Preventing overestimation at these states:* Once we have detected edge-of-reach states, we may simply apply value pessimism methods from the offline model-free literature. Our choice of an ensemble for part *(A)* conveniently allows us to minimize over an ensemble of $Q$-functions, which effectively adds value pessimism based on ensemble variance.

Our resulting proposal is Reach-Aware Value Learning (RAVL). Concretely, we take the standard offline model-based RL procedure (see Algorithm 1), and simply exchange the *dynamics* pessimism penalty for *value* pessimism using minimization over an ensemble of $N$ $Q$-functions:

$$Q_\phi^n(s,a) \leftarrow r + \gamma \min_{i=1,\ldots,N} Q_\phi^i(s', \pi_\theta(s')) \quad \text{for } n = 1, \ldots, N \tag{2}$$

We include EDAC's [1] ensemble diversity regularizer, and in Section 7 we discuss how the impact of value pessimism differs significantly in the model-based (RAVL) vs model-free (EDAC) settings.

## 6 Empirical Evaluation

In this section, we begin by analyzing RAVL on the simple environment from Section 4. Next, we look at the standard D4RL benchmark, first confirming that RAVL solves the failure seen with the true dynamics, before then demonstrating that RAVL achieves strong performance with the

learned dynamics. In Appendix F, we include additional results on the challenging pixel-based V-D4RL benchmark on which RAVL now represents a new state-of-the-art. Finally in Section 6.5 we reexamine prior model-based methods and explain why they may work despite not explicitly addressing the edge-of-reach problem. We provide full hyperparameters in Appendix C.2 and ablations showing that RAVL is stable over hyperparameter choice in Appendix G.2.

## 6.1 Simple Environment

Testing on the simple environment from Section 4 (see **green** Figure 4), we observe that RAVL behaves the same as the theoretically optimal but practically impossible *Base-OraclePatch* method. Moreover, in Figure 5, we see that the $Q$-value variance over the ensemble is significantly higher for edge-of-reach states, meaning RAVL is detecting and penalizing edge-of-reach states exactly as intended.

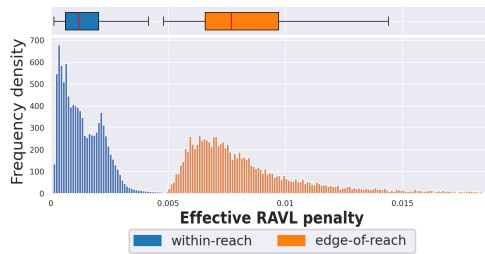

Figure 5: RAVL's effective penalty of $Q$-ensemble variance on the environment in Section 4, showing that - as intended - edge-of-reach states have significantly higher penalty than within-reach states.

## 6.2 D4RL with the True Dynamics

Next, we demonstrate that RAVL works without dynamics uncertainty and solves the 'failure' observed in Section 3.1. Table 1 shows results on the standard offline benchmark D4RL [7] MuJoCo [34] v2 datasets with the true (zero error, zero uncertainty) dynamics. We see that RAVL learns the near-optimal policies, while existing methods using the base model-based procedure (Algorithm 1) completely fail. In Section 6.5 we examine how this is because existing methods overlook the critical edge-of-reach problem and instead only accidentally address it using dynamics uncertainty metrics. In the absence of model uncertainty, these methods have no correction for edge-of-reach states and hence fail dramatically.

Table 1: **True dynamics (zero error, zero uncertainty)** Existing model-based methods are presented as different approaches for dealing with dynamics model errors. Surprisingly, however, all existing methods fail in the absence of dynamics errors (when the learned **approximate** model is replaced by the **true** model). This reveals that existing methods are unintentionally using their dynamics uncertainty estimates to address the previously unnoticed edge-of-reach problem. By contrast, RAVL directly addresses the edge-of-reach problem and hence does not fail when dynamics uncertainty is zero. Experiments are on the D4RL MuJoCo v2 datasets. Statistical significance highlighted (6 seeds). **\*\*Note that while labeled as 'MOBILE', the results with the true dynamics will be identical for any other dynamics penalty-based method since penalties under the true model are all zero (see Table 7). Hence these results indicate the failure of all existing dynamics uncertainty-based methods.**

|  | | MOBILE (SOTA)\*\* | | RAVL | |
| --- | --- | --- | --- | --- | --- |
| **Dynamics model** | | **Approximate** | **True** | **Approximate** | **True** |
| Halfcheetah | medium | 74.6±1.2 | 72.2±4.1 (↓**7%**) | 78.7±2.0 | 76.4±3.4 |
| | mixed | 71.7±1.2 | 72.2±4.1 (↑**1%**) | 74.9±2.0 | 77.3±2.2 |
| | medexp | 108.2±2.5 | 84.9±17.6 (↓**22%**) | 102.1±8.0 | 109.3±1.3 |
| Hopper | medium | 106.6±0.6 | **17.2±11.1** (↓**84%**) | 90.6±11.9 | 101.6±0.9 |
| | mixed | 103.9±1.0 | 71.5±32.2 (↓**31%**) | 103.1±1.1 | 101.1±0.5 |
| | medexp | 112.6±0.2 | **6.1±6.8** (↓**96%**) | 110.1±1.8 | 107.2±0.9 |
| Walker2d | medium | 87.7±1.1 | **7.2±1.6** (↓**92%**) | 86.3±1.6 | 101.2±2.3 |
| | mixed | 89.9±1.5 | **7.9±1.9** (↓**91%**) | 83.0±2.5 | 86.4±3.0 |
| | medexp | 115.2±0.7 | **7.7±2.1** (↓**93%**) | 115.5±2.4 | 103.5±1.2 |

## 6.3 D4RL with Learned Dynamics

Next, we show that RAVL also performs well with a learned dynamics model. Figure 3 shows RAVL successfully stabilizes the $Q$-value explosion of the base procedure, and Table 2 shows RAVL largely matches SOTA, while having *significantly lower runtime* (see Section 6.4). RAVL gives much higher performance on the Halfcheetah mixed and medium datasets than its model-free counterpart EDAC, and in Section 6.5, we discuss why we would expect the effect of value pessimism in the model-based setting (RAVL) to inherently offer much more flexibility than in the model-free setting (EDAC).

Table 2: A comprehensive evaluation of RAVL over the standard D4RL MuJoCo benchmark. We show the mean and standard deviation of the final performance averaged over 6 seeds. Our simple approach largely matches the state-of-the-art without any explicit dynamics penalization and hence works even in the absence of model uncertainty (where dynamics uncertainty-based methods fail) (see Table 1).

| Environment | | Model-Free | | | Model-Based | | | | |
| --- | --- | --- | --- | --- | --- | --- | --- | --- | --- |
| | | BC | CQL | EDAC | MOPO | COMBO | RAMBO | MOBILE | RAVL (Ours) |
| Halfcheetah | random | 2.2 | 31.3 | 28.4 | 38.5 | 38.8 | 39.5 | **39.3** | 34.4±2.0 |
| | medium | 43.2 | 46.9 | 65.9 | 73.0 | 54.2 | 77.9 | 74.6 | **78.7±2.0** |
| | mixed | 37.6 | 45.3 | 61.3 | 72.1 | 55.1 | 68.7 | 71.7 | **74.9±2.0** |
| | medexp | 44.0 | 95.0 | **106.3** | 90.8 | 90.0 | 95.4 | 108.2 | 102.1±8.0 |
| Hopper | random | 3.7 | 5.3 | 25.3 | 31.7 | 17.9 | 25.4 | **31.9** | 31.4±0.1 |
| | medium | 54.1 | 61.9 | 101.6 | 62.8 | 97.2 | 87.0 | **106.6** | 90.6±11.9 |
| | mixed | 16.6 | 86.3 | 101.0 | 103.5 | 89.5 | 99.5 | 103.9 | **103.1±1.1** |
| | medexp | 53.9 | 96.9 | 110.7 | 81.6 | 111.1 | 88.2 | **112.6** | 110.1±1.8 |
| Walker2d | random | 1.3 | 5.4 | 16.6 | 7.4 | 7.0 | 0.0 | **17.9** | 17.1±7.9 |
| | medium | 70.9 | 79.5 | **92.5** | 84.1 | 81.9 | 84.9 | 87.7 | 86.3±1.6 |
| | mixed | 20.3 | 76.8 | 87.1 | 85.6 | 56.0 | 89.2 | **89.9** | 83.0±2.5 |
| | medexp | 90.1 | 109.1 | **114.7** | 112.9 | 103.3 | 56.7 | 115.2 | 115.5±2.4 |

We additionally include results for the challenging *pixel-based* **V-D4RL benchmark** for which *latent-space models* are used (in Appendix F), and accompanying **ablation experiments** (in Appendix G.2). In this setting, RAVL represents a new SOTA, giving a performance boost of more than 20% for some environments. These results are particularly notable as the pixel-based setting means the base algorithm (DreamerV2) uses model rollouts in an imagined latent space (rather than the original state-action space as in MBPO). The results therefore give promising evidence that RAVL is able to generalize well to different representation spaces.

## 6.4 Runtime Discussion

Vectorized ensembles can be scaled with extremely minimal effect on the runtime (see Table 10). This means that, *per epoch*, RAVL is approximately 13% faster than the SOTA (MOBILE, due to MOBILE needing multiple extra forward passes to compute its uncertainty penalty). Furthermore, *in total*, we find that RAVL reliably requires $3\times$ fewer epochs to converge on all but the medexp datasets, meaning the total runtime is approximately 70% faster than SOTA (see Appendix G.1).

## 6.5 How can prior methods work despite overlooking the edge-of-reach problem?

While ostensibly to address model errors, we find that existing dynamics penalties *accidentally* address the edge-of-reach problem: In Figure 6 we see a positive correlation between the penalties used in dynamics uncertainty methods and RAVL's effective penalty of value ensemble variance. This may be expected, as dynamics uncertainty will naturally be higher further away from $\mathcal{D}_{\text{offline}}$, which is also where edge-of-reach states are more likely to lie. In Section 3.4 we present evidence suggesting that the edge-of-reach problem is likely the dominant source of issues on the main D4RL benchmark, thus indicating that the dynamics uncertainty penalties of existing methods are likely indirectly addressing the edge-of-reach problem. In general, dynamics errors are a second orthogonal problem and RAVL can be easily combined with appropriate dynamics uncertainty penalization [24, 29, 36] for environments where this is a significant issue.

## 7 Related Work

**Model-Based Methods.** Existing offline model-based methods present dynamics model errors and consequent model exploitation as the sole source of issues. A broad class of methods therefore propose reward penalties based on the estimated level of model uncertainty [2, 14, 24, 29, 36], typically using variance over a dynamics ensemble. Rigter et al. [27] aim to avoid model exploitation by setting up an adversarial two-player game between the policy and model. Finally, most related to our method, COMBO [37], penalizes value estimates for state-actions outside model rollouts. However, similarly to Yu et al. [36], COMBO is theoretically motivated by the assumption of infinite horizon model rollouts, which we show overlooks serious implications. Critically, in contrast to our approach, none of these methods address the edge-of-reach problem and thus they fail as environment

models become more accurate. A related phenomenon of overestimation stemming from hallucinated states has been observed in online model-based RL[12].

**Offline Model-Free Methods.** Model-free methods can broadly be divided into two approaches to solving the out-of-sample action problem central to offline model-free RL (see Section 2): action constraint methods and value pessimism-based methods. Action constraint methods [8, 17, 18] aim to avoid using out-of-sample actions in the Bellman update by ensuring selected actions are close to the dataset behavior policy $\pi^\beta$. By contrast, value pessimism-based methods aim to directly regularize the value function to produce low-value estimates for out-of-sample state-actions [1, 16, 19]. The *edge-of-reach* problem is the model-based equivalent of the *out-of-sample* action problem, and this unified understanding allows us to transfer ideas directly from model-free literature. RAVL is based on EDAC's [1] use of minimization over a $Q$-ensemble [9, 35], and applies this to model-based offline RL.

### *What is the effect of value pessimism in the model-based vs model-free settings?*

EDAC [1] can be seen as RAVL's model-free counterpart, however, the impact of value pessimism in the model-free vs the model-based settings is notably different. Recall that, with the ensemble $Q$-function trained on $(s, a, r, s') \sim \mathcal{D}_\square$, the state-actions that are penalized (due to being outside the training distribution) are any $(s', a')$ that are out-of-distribution with respect to the $(s, a)$'s in the dataset $\mathcal{D}_\square$. Recall also that updates use values at $(s', a')$ where $a' \sim \pi_\theta$ (see Equation (1)).

*In the model-free case (EDAC):* The dataset is $\mathcal{D}_\square = \mathcal{D}_{offline}$, meaning the actions $a$ are effectively sampled from the dataset behavior policy (off-policy), whereas the actions $a'$ are sampled on-policy. This means that EDAC penalizes any $(s', a')$ where $a'$ differs significantly from the behavior policy.

*In the model-based case (RAVL):* The dataset is $\mathcal{D}_\square = \mathcal{D}_{rollouts}$, meaning now both $a$ and $a'$ are sampled on-policy. As a result, the only $(s', a')$ which will now be out-of-distribution with respect to $(s, a)$ (and hence penalized) are those where the state $s'$ is out-of-distribution with respect to the states $s$ in $\mathcal{D}_{rollouts}$. This happens when $s'$ is reachable only in the final step of rollouts, i.e. exactly when $s'$ is "edge-of-reach" (as illustrated in Figure 2). Compared to EDAC, RAVL can therefore be viewed as "relaxing" the penalty and giving the agent *freedom to learn a policy that differs significantly from the dataset behavior policy*. This distinction is covered in detail in Appendix A.

## 8 Conclusion

This paper investigates how offline model-based methods perform as dynamics models become more accurate. As an interesting hypothetical extreme, we test existing methods with the true error-free dynamics. Surprisingly, we find that all existing methods fail. This reveals that using truncated rollout horizons (as per the shared base procedure) has critical and previously overlooked consequences stemming from the consequent existence of 'edge-of-reach' states. We show that existing methods are *indirectly* and *accidentally* addressing this edge-of-reach problem (rather than addressing model errors as stated), and hence explain why they fail catastrophically with the true dynamics.

This problem reveals close connections between model-based and model-free approaches and leads us to present a unified perspective for offline RL. Based on this, we propose RAVL, a simple and robust method that achieves strong performance across both proprioceptive and pixel-based benchmarks. Moreover, RAVL *directly* addresses the edge-of-reach problem, meaning that - unlike existing methods - RAVL does not fail under the true environment model, and has the practical benefit of not requiring dynamics uncertainty estimates. Since improvements to dynamics models are inevitable, we believe that resolving the brittle and unanticipated failure of existing methods under dynamics model improvements is an important step towards 'future-proofing' offline RL.

## 9 Limitations

Since dynamics models for the main offline RL benchmarks are highly accurate, the edge-of-reach effects dominate, and RAVL is sufficient to stabilize model-based training effectively without any explicit dynamics uncertainty penalty. In general, however, edge-of-reach issues could be mixed with dynamics error, and understanding how to balance these two concerns would be useful future work. Further, we believe that studying the impact of the edge-of-reach effect in a wider setting could be an exciting direction, for example investigating its effect as an implicit exploration bias in online RL.

## Acknowledgments and Disclosure of Funding

Anya Sims and Cong Lu are funded by the Engineering and Physical Sciences Research Council (EPSRC). The authors would like to thank the anonymous Agent Learning in Open-Endedness Workshop at NeurIPS 2023 and ICLR 2024 reviewers for positive and constructive feedback which helped to improve the paper. We would also like to thank Philip J. Ball and Shimon Whiteson for helpful feedback and discussions on earlier drafts of this work.

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

# Supplementary Material

## Table of Contents

# A A Unified Perspective of Model-Based and Model-Free RL

We supplement the discussion in Section 3.2 with a more thorough comparison of the out-of-sample and edge-of-reach problems, including how they relate to model-free and model-based approaches.

## A.1 Definitions

Consider a dataset of transition tuples $\mathcal{D} = \{(s_i, a_i, r_i, s_i', d_i)\}_{i=1,...,N}$ collected according to some dataset policy $\pi^{\mathcal{D}}(\cdot|s)$. Compared to Section 2, we include the addition of a *done* indicator $d_i$, where $d_i = 1$ indicates episode termination (and $d_i = 0$ otherwise). Transition tuples thus consist of *state*, *action*, *reward*, *nextstate*, *done*. Consider the marginal distribution over state-actions $\rho_{s,a}^{\mathcal{D}}(\cdot, \cdot)$, over states $\rho_s^{\mathcal{D}}(\cdot)$, and conditional action distribution $\rho_{a|s}^{\mathcal{D}}(\cdot|s)$. Note that $\rho_{a|s}^{\mathcal{D}}(\cdot|s) = \pi^{\mathcal{D}}(\cdot|s)$. We abbreviate *x is in distribution with respect to $\rho$* as $x \in^{\text{dist}} \rho$.

## A.2 Q-Learning Conditions

As described in Section 2, given some policy $\pi$, we can attempt to learn the corresponding $Q$-function with the following iterative process:

$$Q^{k+1} \leftarrow \arg\min_Q \mathbb{E}_{(s,a,r,s')\sim\mathcal{D}, a'\sim\pi(\cdot|s')}[(\underbrace{Q(s,a)}_{\text{input}} - \underbrace{[r + \gamma(1-d)Q^k(s',a')]}_{\text{Bellman target}})^2] \qquad (3)$$

$Q$-learning relies on bootstrapping, hence to be successful we need to be able to learn accurate estimates of the Bellman targets for all $(s, a)$ inputs. Bootstrapped estimates of $Q(s', a')$ are used in the targets whenever $d \neq 1$. Therefore, for all $(s', a')$, we require:

**Combined state-action condition**: $(s', a') \in^{\text{dist}} \rho_{s,a}^{\mathcal{D}}$ or $d = 1$.

In the main paper, we use this combined state-action perspective for simplicity, however, we can equivalently divide this state-action condition into independent requirements on the state and action as follows:

**State condition**: $s' \in^{\text{dist}} \rho_s^{\mathcal{D}}$ or $d = 1$,

**Action condition**: $a' \in^{\text{dist}} \rho_{a|s}^{\mathcal{D}}(s')$ (given the above condition is met and $d \neq 1$).

Informally, the state condition may be violated if $\mathcal{D}$ consists of partial or truncated trajectories, and the action condition may be violated if there is a significant distribution shift between $\pi^{\mathcal{D}}$ and $\pi$.

## A.3 Comparison Between Model-Free and Model-Based Methods

In offline model-free RL, $\mathcal{D} = \mathcal{D}_{\text{offline}}$, with $\pi^{\mathcal{D}} = \pi^{\beta}$. For the settings we consider, $\mathcal{D}_{\text{offline}}$ consists of full trajectories and therefore will not violate the state condition. However, this may happen in a more general setting with $\mathcal{D}_{\text{offline}}$ containing truncated trajectories. By contrast, the mismatch between $\pi$ (used to sample $a'$ in $Q$-learning) and $\pi^{\beta}$ (used to sample $a$ in the dataset $\mathcal{D}_{\text{offline}}$) often does lead to **significant violation of the action condition**. This exacerbates the overestimation bias in $Q$-learning (see Section 7), and can result in pathological training dynamics and $Q$-value explosion over training [18].

On the other hand, in offline model-based RL, the dataset $\mathcal{D} = \mathcal{D}_{\text{rollouts}}$ is collected *on-policy* according to the current (or recent) policy such that $\pi^{\mathcal{D}} \approx \pi$. This minimal mismatch between $\pi^{\mathcal{D}}$ and $\pi$ means the action condition is not violated and can be considered to be resolved due to the collection of additional data. However, the procedure of generating the data $\mathcal{D} = \mathcal{D}_{\text{rollouts}}$ can be seen to significantly exacerbate the state condition problem, as the use of short truncated-horizon trajectories means the resulting dataset $\mathcal{D}_{\text{rollouts}}$ is **likely to violate the state condition**. Due to lack of exploration, certain states may temporarily violate the state condition. Our paper then considers the pathological case of *edge-of-reach* states, which will always violate the state condition.

This comparison between model-based and model-free is summarized in Table 3

Table 3: A summary of the comparison between model-free and model-based offline RL in relation to the conditions on $Q$-learning as described in Appendix A.

|  | Action condition violation | State condition violation | Main source of issues |
|---|---|---|---|
| **Model-free** | Yes, common in practice | Yes, but uncommon in practice | $\rightarrow$ Action condition violation |
| **Model-based** | No | Yes, common in practice | $\rightarrow$ State condition violation |

# B Error Propagation Result

**Proposition 1** (Error propagation from edge-of-reach states). *Consider a rollout of length $k$, $(s_0, a_0, s_1, \ldots, s_k)$. Suppose that the state $s_k$ is edge-of-reach and the approximate value function $Q^j(s_k, \pi(s_k))$ has error $\epsilon$. Then, standard value iteration will compound error $\gamma^{k-t}\epsilon$ to the estimates of $Q^{j+1}(s_t, a_t)$ for $t = 1, \ldots, k-1$. (Proof in Appendix B.)*

## B.1 Proof

In this section, we provide a proof of Proposition 1. Our proof follows analogous logic to the error propagation result of Kumar et al. [18].

*Proof.* Let us denote $Q^*$ as the optimal value function, $\zeta_j(s, a) = |Q_j(s, a) - Q^*(s, a)|$ the error at iteration $j$ of Q-Learning, and $\delta_j(s, a) = |Q_j(s, a) - \mathcal{T}Q_{j-1}(s, a)|$ the current Bellman error. Then first considering the $t = k - 1$ case,

$$\zeta_j(s_t, a_t) = |Q_j(s_t, a_t) - Q^*(s_t, a_t)| \tag{4}$$

$$= |Q_j(s_t, a_t) - \mathcal{T}Q_{j-1}(s_t, a_t) + \mathcal{T}Q_{j-1}(s_t, a_t) - Q^*(s_t, a_t)| \tag{5}$$

$$\leq |Q_j(s_t, a_t) - \mathcal{T}Q_{j-1}(s_t, a_t)| + |\mathcal{T}Q_{j-1}(s_t, a_t) - Q^*(s_t, a_t)| \tag{6}$$

$$= \delta_j(s_t, a_t) + \gamma\zeta_{j-1}(s_{t+1}, a_{t+1}) \tag{7}$$

$$= \delta_j(s_t, a_t) + \gamma\epsilon \tag{8}$$

Thus the errors at edge-of-reach states are discounted and then compounded with new errors at $Q^j(s_{k-1}, a_{k-1})$. For $t < k - 1$, the result follows from repeated application of Equation (7) along the rollout.

$\square$

## B.2 Extension to Stochastic Environments and Practical Implementation

With stochastic transition models (e.g. Gaussian models), we may have the case that no state will truly have zero density, in which case we relax the definition of edge-of-reach states slightly (see Section 3.3) from $\rho_{t,\pi}(s) = 0$ to $\rho_{t,\pi}(s) < \epsilon$ for some small $\epsilon$.

During optimization, in practice, model rollouts are sampled in minibatches and thus the above error propagation effect will occur on average throughout training. An analogous model-free statement has been given in Kumar et al. [18]; however, its significance in the context of model-based methods was previously not considered.

# C Implementation Details

In this section, we provide full implementation details for RAVL.

## C.1 Algorithm

We use the base model-based procedure as given in Algorithm 1 and shared across model-based offline RL methods [14, 24, 29, 36]. This involves using short MBPO-style [13] model rollouts to train an agent based on SAC [11]. We modify the SAC agent with the value pessimism losses of EDAC [1]. Our dynamics model follows the standard setup in model-based offline algorithms, being realized as a deep ensemble [5] and trained via maximum likelihood estimation.

## C.2  Hyperparameters

For the D4RL [7] MuJoCo results presented in Table 2, we sweep over the following hyperparameters and list the choices used in Table 4. "Base" refers to the shared base procedure in model-based offline RL shown in Algorithm 1.

- **(EDAC)** Number of $Q$-ensemble elements $N_{\text{critic}}$, in the range $\{10, 50\}$
- **(EDAC)** Ensemble diversity weight $\eta$, in the range $\{1, 10, 100\}$
- **(Base)** Model rollout length $k$, in the range $\{1, 5\}$
- **(Base)** Real-to-synthetic data ratio $r$, in the range $\{0.05, 0.5\}$

The remaining model-based and agent hyperparameters are given in Table 5. Almost all environments use a small $N_{\text{critic}} = 10$, with only the Hopper-medexp dataset needing $N_{\text{critic}} = 30$.

For the uncertainty-free dynamics model experiments in Table 1 we take the same hyperparameters as for the learned dynamics model for $N$, $k$, and $r$, and try different settings for $\eta \in \{1, 10, 100, 200\}$. The choices used are shown in Table 4. Note that for existing methods based on dynamics penalties, the analogous hyperparameter (penalty weighting) has no effect under the true dynamics model as the penalties will always be zero (since dynamics uncertainty is zero). For the experiments in Figure 1 with intermediate dynamics model accuracies, we tune the ensemble diversity coefficient $\eta$ over $\eta \in \{1, 10, 100, 200\}$ for RAVL, and analogously we tune the uncertainty penalty coefficient $\lambda$ over $\lambda \in \{1, 5, 10, 100, 200\}$ for MOPO.

Our implementation is based on the Clean Offline Reinforcement Learning (CORL, Tarasov et al. [33]) repository, released at `https://github.com/tinkoff-ai/CORL` under an Apache-2.0 license. Our algorithm takes on average 6 hours to run using a V100 GPU for the full number of epochs.

Table 4: Variable hyperparameters for RAVL used in D4RL MuJoCo locomotion tasks with the learned dynamics (see Table 2). With the uncertainty-free dynamics (see Table 1), we tune only $\eta$, with the settings used shown in brackets.

| Environment | | $N_{\text{critic}}$ | $\eta$ | k | r |
|---|---|---|---|---|---|
| HalfCheetah | random | 10 | 10 (1) | 5 | 0.5 |
| | medium | 10 | 1 (1) | 5 | 0.05 |
| | mixed | 10 | 100 (1) | 5 | 0.05 |
| | medexp | 10 | 1 (1) | 5 | 0.5 |
| Hopper | random | 10 | 100 (1) | 5 | 0.05 |
| | medium | 10 | 100 (10) | 1 | 0.5 |
| | mixed | 10 | 10 (10) | 1 | 0.5 |
| | medexp | 30 | 100 (10) | 1 | 0.5 |
| Walker2d | random | 10 | 1 (100) | 5 | 0.05 |
| | medium | 10 | 10 (100) | 1 | 0.5 |
| | mixed | 10 | 10 (100) | 1 | 0.5 |
| | medexp | 10 | 1 (200) | 1 | 0.5 |

Table 5: Fixed hyperparameters for RAVL used in D4RL MuJoCo locomotion tasks.

| Parameter | Value |
|---|---|
| epochs | 3,000 for medexp; 1,000 for rest |
| gamma | 0.99 |
| learning rate | $3 \times 10^{-4}$ |
| batch size | 256 |
| buffer retain epochs | 5 |
| number of rollouts | 50,000 |

## C.3  Pixel-Based Hyperparameters

For the V-D4RL [25] DeepMind Control Suite datasets presented in Table 9, we use the default hyperparameters for the Offline DV2 algorithm, which are given in Table 6. We found keeping the

uncertainty weight $\lambda = 10$ improved performance over $\lambda = 0$ which shows RAVL can be combined with dynamics penalty-based methods.

Table 6: Hyperparameters for RAVL used in V-D4RL DeepMind Control Suite tasks.

| Parameter | Value |
|---|---|
| ensemble member count (K) | 7 |
| imagination horizon (H) | 5 |
| batch size | 64 |
| sequence length (L) | 50 |
| action repeat | 2 |
| observation size | [64, 64] |
| discount ($\gamma$) | 0.99 |
| optimizer | Adam |
| learning rate | {model $= 3 \times 10^{-4}$, actor-critic $= 8 \times 10^{-5}$} |
| model training epochs | 800 |
| agent training epochs | 2,400 |
| uncertainty penalty | mean disagreement |
| uncertainty weight ($\lambda$) | 10 |
| number of critics ($N_{\text{critic}}$) | 5 |

We used a hyperparameter sweep over $\{2, 5, 20\}$ for $N_{\text{critic}}$ but found a single value of 5 worked well for all environments we consider.

# D   Additional Tables

## D.1   Existing Methods' Penalties All Collapse to Zero Under the True Dynamics

For completeness, in Table 7 we include the expressions for the various dynamics penalties proposed by the most popular model-based approaches. These penalties are all based on estimates of dynamics model uncertainty. Since there is no uncertainty under the true model, all these penalties collapse to zero with the true dynamics.

Table 7: We show here how all existing dynamics penalized offline MBRL algorithms reduce to the same base procedure when estimated (epistemic) model uncertainty is zero (as it is with the true dynamics).

| Offline MBRL Algorithm | Penalty | Penalty with the true dynamics |
|---|---|---|
| MOPO [36] | $\max_{i=1,\ldots,N} \lVert \Sigma_\phi^i(s,a) \rVert_{\text{F}}$ | 0 |
| MOReL [14] | $\max_{1 \leq i,j \leq N} \lVert \mu_\phi^i(s,a) - \mu_\phi^j(s,a) \rVert_2$ | 0 |
| MOBILE [29] | $\text{Std}_{i=1,\ldots,N}\{\hat{T}^i(s,a)\}$ | 0 |

## D.2   Model Errors Cannot Explain the $Q$-value Overestimation

In Figure 3 we observe that the base offline model-based procedure results in $Q$-values exploding unboundedly to beyond $10^{10}$ over training. Existing methods attribute this to dynamics model errors and the consequent exploitation of state-action space areas where the model incorrectly predicts high reward.

In Table 8 we examine the rewards collected by an agent in the learned dynamics model. Under the 'model errors' explanation, we would expect to see rewards sampled by the agent in the learned model to be significantly higher than under the true environment. Furthermore, to explain the $Q$-values being on the order of $10^{10}$, with $\gamma = 0.99$ the agent would need to sample rewards on the order of $10^8$. However, the per-step rewards sampled by the agent are on the order of $10^0$ (and are not higher than with the true environment).

Model errors and model exploitation therefore cannot explain the explosion in $Q$-values seen over training. By contrast, the edge-of-reach problem exactly predicts this extreme value overestimation behavior.

Table 8: Per-step rewards with the base offline model-based procedure (see Algorithm 1). Rewards in model rollouts are close to those with the true dynamics, showing that model exploitation could not explain the subsequent value overestimation. Further, it indicates (as in Janner et al. [13], Lu et al. [24]) that the model is largely accurate for short rollouts and hence is unlikely to be vulnerable to model exploitation.

| Environment | Model reward | True reward |
|---|---|---|
| Hc-med | 4.89±1.27 | 5.01±1.09 |
| Hop-mixed | 2.37±1.05 | 2.44±1.02 |
| W2d-medexp | 3.96±1.41 | 3.96±1.41 |

# E    Additional Visualizations

## E.1    Comparison to Prior Approaches

In Figure 6, we plot the dynamics uncertainty-based penalty used in MOPO [36] against the effective penalty in RAVL of variance of the value ensemble. The positive correlation between MOPO's penalty and the edge-of-reach states targeting penalty of RAVL indicates why prior methods may work despite not considering the crucial edge-of-reach problem.

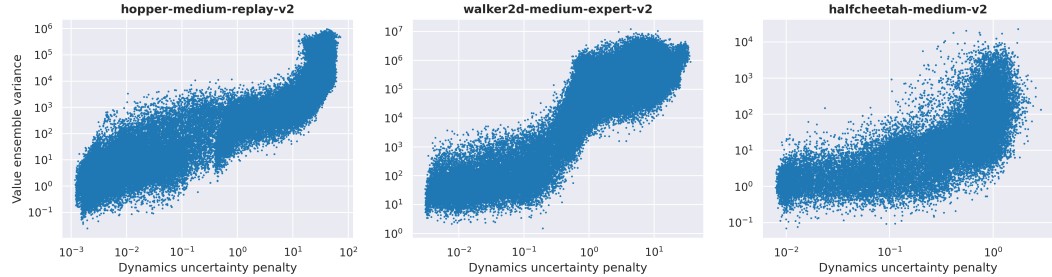

Figure 6: We find that the dynamics uncertainty-based penalty used in MOPO [36] is positively correlated with the variance of the value ensemble of RAVL, suggesting prior methods may unintentionally address the edge-of-reach problem. Pearson correlation coefficients are 0.49, 0.43, and 0.27 for Hopper-mixed, Walker2d-medexp, and Halfcheetah-medium respectively.

## E.2    Behaviour of Rollouts Over Training

In Figure 7, we provide a visualization of the rollouts sampled over training in the simple environment for each of the algorithms analyzed in Figure 4 (see Section 4). This accompanies the discussion in Section 4.2 of the behavior over training.

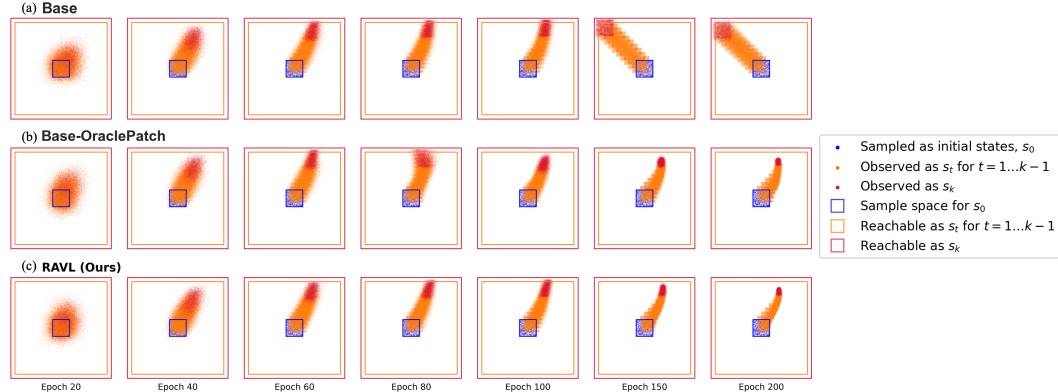

Figure 7: A visualization of the rollouts sampled over training on the simple environment in Section 4. We note the pathological behavior of the baseline, and then the success of the ideal intervention Base-OraclePatch, and our practically realizable method RAVL.

## F    Evaluation on the Pixel-Based V-D4RL Benchmark

The insights from RAVL also lead to improvements in performance on the challenging pixel-based benchmark V-D4RL [25]. The base procedure in this pixel-based setting uses the Offline DreamerV2 algorithm of training on trajectories in a *learned latent space*.

We observe in Table 9 that RAVL gives a strong boost in performance on the medium and medexp level datasets, while helping less in the more diverse random and mixed datasets. This observation fits with our intuition of the edge-of-reach problem: the medium and medexp level datasets are likely to have less coverage of the state space, and thus we would expect them to suffer more from edge-of-reach issues and the 'edge-of-reach seeking behavior' demonstrated in Section 4.2. The performance improvements over Offline DreamerV2 and other model-based baselines including LOMPO [26] therefore suggests that the edge-of-reach problem is general and widespread.

We also note that these results are without the ensemble diversity regularizer from EDAC [1] (used on the D4RL benchmark), which we anticipate may further increase performance.

Table 9: We show that RAVL extends to the challenging pixel-based V-D4RL benchmark, suggesting the out-of-reach problem is also present in the latent-space setting used by the base procedure for pixel-based algorithms. Mean and standard deviation given over 6 seeds.

| V-D4RL Environment | | Offline DV2 | LOMPO | RAVL (Ours) |
|---|---|---|---|---|
| Walker-walk | random | **28.7**±**13.0** | 21.9±8.1 | **29.7**±**2.4** |
| | mixed | **56.5**±**18.1** | 34.7±19.7 | **49.3**±**11.3** |
| | medium | 34.1±19.7 | 43.4±11.1 | **48.6**±**8.7** |
| | medexp | **43.9**±**34.4** | 39.2±19.5 | **47.6**±**18.3** |
| Cheetah-run | random | 31.7±2.7 | 11.4±5.1 | **36.9**±**5.0** |
| | mixed | **61.6**±**1.0** | 36.3±13.6 | 57.6±1.4 |
| | medium | **17.2**±**3.5** | 16.4±8.3 | **18.3**±**4.3** |
| | medexp | **10.4**±**3.5** | 11.9±1.9 | **12.6**±**4.4** |
| Overall | | **35.5**±**12.0** | 26.9±10.9 | **37.6**±**7.0** |

## G    Runtime and Hyperparameter Sensitivity Ablations

### G.1    Runtime

We compare RAVL to SOTA method MOBILE. The additional requirement for *RAVL compared to MOBILE* is that RAVL requires increasing the $Q$-ensemble size: from $N = 2$ in MOBILE to $N = 10$ in RAVL (for all environments except one where we use $N = 30$). The additional requirement for *MOBILE compared to RAVL* is that MOBILE requires multiple additional forwards passes through the model for each update step (for computing their dynamics uncertainty penalty).

In Table 10 we show that ensemble size can be scaled with very minimal effect on the runtime. Even increasing the ensemble size to $N = 100$ (far beyond the maximum ensemble size used by RAVL) only increases the runtime by at most 1%. Table 7 of MOBILE reports that MOBILE's requirement of additional forward passes increases runtime by around 14%.[6]

Table 10: Timing experiments showing that ensemble size for standard vectorized ensembles can be scaled with extremely minimal effect on runtime. Runtime increases by just 1% with $Q$-ensemble size increased from $N = 2$ (as used in the base procedure SAC agent) to $N = 100$. RAVL uses $N = 10$ (for all environments except one where we use $N = 30$), meaning the runtime increase compared to the base model-based procedure is almost negligible. Other methods add more computationally expensive changes to the base procedure, hence making RAVL significantly faster than SOTA (see Appendix G.1).

| Ensemble size ($N$) | 2 | 10 | 50 | 100 |
|---|---|---|---|---|
| Time relative to $N = 2$ | (baseline) | $\times 1.00 \pm 0.03$ | $\times 1.01 \pm 0.04$ | $\times 1.01 \pm 0.02$ |

## G.2 Hyperparameter Sensitivity Ablations

In Table 11 we include results with different settings of RAVL's ensmble diversity regularizer hyperparameter $\eta$. We note a minimal change in performance with a sweep across two orders of magnitude. This is an indication that RAVL may be applied to new settings without much tuning.

For RAVL's second hyperparameter of $Q$-ensemble size $N$, we note that in the main benchmarking of RAVL against other offline methods (see Table 2) we use the same value of $N = 10$ across all except one environment. This again indicates that RAVL is robust and should transfer to new settings with minimal hyperparameter tuning.

Table 11: Ablations results over RAVL's diversity regularizer hyperparameter $\eta$.

| Hyperparameter $\eta$ | Halfcheetah medium | Halfcheetah mixed |
|---|---|---|
| 1.0 | $78.7 \pm 2.0$ | $61.7 \pm 5.6$ |
| 10.0 | $74.8 \pm 1.5$ | $73.8 \pm 1.9$ |
| 100.0 | $72.1 \pm 2.2$ | $74.9 \pm 2.0$ |

## H Summary of Setups Used in Comparisons

Throughout the paper, we compare several different setups in order to identify the true underlying issues in model-based offline RL. We provide a summary of them in Table 12. More comprehensive descriptions of each are given in the relevant positions in the main text and table and figure captions.

Table 12: We summarize the various setups used for comparisons throughout the paper. '*' denotes application to the simple environment (see Section 4). All methods use $k$-step rollouts from the offline dataset (or from a fixed starting state distribution in the case of the simple environment).

| *Figures and Tables* | Name Used | Dynamics Model | | Agent | |
|---|---|---|---|---|---|
| | | Type | Penalty | $N_{\mathbf{critic}}$ | $\eta$ |
| *Table* 1, *Figure* 6 | **'Approximate'** | Ensemble | ✓(Best of Table 7) | 2 | 0 |
| *Table* 1 | **'True'** | True | n/a | 2 | 0 |
| *Tables* 8, *Figure* 3 | **'Base'** | Ensemble | ✗ | 2 | 0 |
| *Tables* 8, 2, *Figure* 3 | **'RAVL (Ours)'** | Ensemble | ✗ | $> 2$ | $> 0$ |
| *Figures* 4, 7 | **'Base'\*** | True | n/a | 2 | 0 |
| *Figures* 4, 5, 7 | **'RAVL (Ours)'\*** | True | n/a | $> 2$ | $> 0$ |

---

[6]Note that the times reported for EDAC in Table 7 of MOBILE are not representative of practice, since these are with a non-vectorized $Q$-ensemble (confirmed from their open-source code), which would require just a simple change to 5 lines of code.

