# OpenReview forum: "The Edge-of-Reach Problem in Offline Model-Based Reinforcement Learning"
_NeurIPS.cc/2024/Conference — NeurIPS 2024 poster_

### Official Review · Reviewer_8ru8 · 2024-06-28

**Soundness:** 3
**Presentation:** 3
**Contribution:** 3
**Rating:** 7
**Confidence:** 4

**Summary:**

This paper identifies an overlooked problem in offline model-based reinforcement learning (RL).
Offline model-based RL approaches commonly perform online RL inside a learned dynamics model and
due to model errors, they generally use truncated $k$ step rollouts, instead of full trajectories (episodes).
In this paper, the authors replace the learned dynamics model with the true simulator and show that offline model-based
RL approaches fail even when they have access to the true dynamics.
This is due to states which can only be reached at the end of the $k$ step rollouts (which the authors coin "edge-of-reach" states) and thus, only appear in the target of the Q update. As such, the Q-function is never updated at these "edge-of-reach" states so any overestimation is gradually propagated over the entire state-action space.
The authors provide a nice diagram which provides intuition and shows the effect in a simple environment.
They then propose an algorithm which overcomes the so-called "edge-of-reach" problem by using an ensemble of Q functions.
They demonstrate good performance in D4RL and V-D4RL.

**Strengths:**

I think this is a good paper that should be accepted.
It identifies an overlooked problem in offline model-based RL which seems like it will be an important contribution to the community.
The authors have also done a nice job providing intuition for the problem via Figure 2 and the example in Section 4.

Based on the identified problem, the authors also propose a novel method which is simple yet effective.
They have also demonstrated the effectiveness of their approach on data sets with both proprioceptive and image-based observations.

**Weaknesses:**

I have no major weaknesses. Nevertheless, I do have some comments which I provide to help improve the paper.
The main weakness of this paper is that it looks messy and overcrowded. There are multiple reasons for this which I will now detail.

The use of italics throughout the text makes the paper look very messy.
Sometimes the italics refer to questions whilst other times they just seem to be random words.
I would advise the authors to remove all italics.

There are too many subsections.
For example, in Section 6 there are 5 subsections and each subsection is a single paragraph.
I would advise the authors to replace \subsection with \paragraph.
Similarly, on Lines 283/293/303 the authors use unnumbered subsections. This looks very messy.
I suggest the authors replace this with \paragraph so that the text starts immediately after the bold text and does not have a line break before.

There are a lot of incorrect uses of colons, which makes the writing quality feel poor. Please remove the colons on lines 12/90/114/122/137/144/154/176/195/273/305.

The authors do not mention how the Q-function is initialized. I think it is very important to include all of this information.
For example, if we initialize the Q-function to be zero at all inputs wouldn't that help prevent overestimation bias from "edge-of-reach" states?

## Minor comments and typos
- Line 89 - $\\}\_{i=1,\ldots,N}$ should be $\\}\_{i=1}^{N}$
- Line 122 - incorrect use of semicolon. This should just be "and".
- Line 137 - the full stop should be outside the quote.
- Line 167 - $0\ldots k-1$ should be $0,\ldots,k-1$
- Line 170 - Why is the last sentence of the definition not italicised?
- Line 249 - the comma should be outside the quote.
- Line 306/308/309/31 - the dataset subscript looks wrong.
- Sections shouldn't lead straight into subsections, e.g. 6 into 6.1.
    - Add a few sentences explaining what you are introducing in this section.

**Questions:**

- What is on the x-axis of Figure 5? What environment is this from? This is not clear from the text or caption.

**Limitations:**

The limitations are adequately discussed.

---

> ### Author Rebuttal · Authors · 2024-08-07
>
> We would like to thank the reviewer for their obvious extreme care in providing us with feedback. We are encouraged by the reviewer’s comments that the paper “should be accepted,” "will be an important contribution to the community,” and are grateful for their detailed comments which will help us improve our paper.
>
> ---
>
> **Weaknesses - Formatting, punctuation, use of subsections, etc.**
>
> We thank the reviewer for their detailed feedback on the organization and formatting of the paper. Having revisited the paper, we completely agree with the reviewer’s comments. We will use the extra page allowance to make the paper feel less crowded, and will make all of the suggested changes (use of italics, commas and colons, subsections, etc.) ahead of the camera-ready submission. We are grateful for the care taken in providing a detailed list of suggested edits, and would like to thank the reviewer again for their time in helping us improve the paper.
>
> **Q1 - Implementation details of Q-function initialization**
>
> We use the default initialization scheme (Kaiming uniform). We thank the reviewer for asking this and will add this implementation detail to the paper.
>
> **Q2 - Effect of initializing the Q-function at zero**
>
> [Please see the Rebuttal PDF.]
>
> The reviewer asked whether the edge-of-reach problem could be solved by simply initializing the Q-function to be zero for all inputs. This is a good question and gets to the heart of the pathological issues involved in the edge-of-reach problem. Initializing a network so that its output is zero given any input is non-trivial. (The obvious way to do so would be to initialize all weights to zero, but the network would then not train.) Thus, we instead we consider what would happen if we initialize so that all Q-value outputs are very close to zero.
>
> As the reviewer commented, in this case we might expect the edge-of-reach problem to be avoided since edge-of-reach states cannot then be a source of overestimation if their Q-values are significantly less than the optimal Q-values. However, as shown in the histograms in the rebuttal PDF, the standard NN initialization that we use already in all our experiments already results in all Q-values being very close to zero at initialization ($<1e-2$). Moreover, all these values are much lower than the optimal Q-values ($\sim 5e+3$) and much lower that the Q-values reached over training ($>1e+10$) (see Figure 3). Yet we still observe the edge-of-reach problem. This means the above line of reasoning does not hold.
>
> Instead what happens is that, when values at within-reach states are increased, the use of NNs as function approximators means that the Q-values at nearby edge-of-reach states may also increase. Q-values at edge-of-reach states are therefore not guaranteed to remain small over training, and hence can become a source of overestimation. Moreover, overestimated Q-values lead to other nearby Q-values also being raised, which can then lead to the original Q-values being raised further, thus setting up a pathological positive feedback cycle whereby Q-values increase unboundedly over training (as seen in Figure 3). This is investigated and explained more thoroughly in [1] and [2] in the context of the analogous model-free offline problem (where function approximation means out-of-sample state-actions become a source of pathological overestimation).
>
> We thank the reviewer for this important question. We will use the camera-ready extra page allowance to include some more of the detail from [1] on the role function approximation and this pathological positive feedback loop.
>
> [1] - Aviral Kumar, et al. Stabilizing Off-Policy Q-Learning via Bootstrapping Error Reduction, 2019.
>
> [2] - Sergey Levine, et al. Offline reinforcement learning: Tutorial, review, and perspectives on open problems, 2020.
>
> **Q3 - Clarification on the $x$-axis of Figure 5**
>
> The $x$-axis of Figure 5 shows variance over RAVL’s Q-ensemble. Since RAVL takes the minimum over this Q-ensemble (see Equation 2), the penalty being added is effectively proportional to the Q-ensemble variance, hence why we labeled the $x$-axis as “RAVL’s effective penalty.” We described this briefly in the caption, but we will make this clearer in the camera-ready version.
>
> ---
>
> We thank the reviewer again for their careful review of our submission. The question about initializing the Q-function at zero was particularly insightful. We are also extremely grateful for the feedback about formatting and the detailed list of suggested edits. We agree with these and will make the suggested changes. Thank you for your help in improving our paper. Please let us know if you have any more questions or comments as we would be more than happy to discuss further.

---

> > ### Comment · Reviewer_8ru8 · 2024-08-08
> >
> > I would like to thank the authors for considering my comments and providing a detailed rebuttal. I have also taken the time to read the other reviews and comments. I believe this paper should be accepted so I stand with my initial score of 7.
> >
> >
> > One minor comment regarding Q-function initialisation. Could you not just initialise the Q-function by misusing it from itself and freezing the extra network, like,
> > $$Q(s,a) = Q_{\theta}(s,a) - Q_{\theta_0}^{\text{frozen}}(s,a).$$
> > I appreciate this would probably not solve the issue, but this is what I meant when I suggested initialising the Q-function to be zero everywhere.

---

> > > ### Author Response · Authors · 2024-08-10
> > >
> > > Thank you for your reply. We had not considered this initialization. As you said, it would still not be expected to solve the problem as the overestimation mechanism described in our original answer still holds.
> > >
> > > We have run some experiments to confirm this. The table below shows the final performance and $Q$-values for 3 seeds of Walker2D-medium with MBPO with and without this initialization method. As expected, this does not solve the problem and we see the same Q-value explosion and no significant difference between these initialization methods. (For reference, expert performance is $\sim 100$, and correct Q-values of the optimal policy are $\sim 5e+3$.)
> > >
> > > |    *(3 seeds)*  |  Final Score |  Final $Q$-values  |
> > > |:-:|:-:|:-:|
> > > |Default initialization       | (-0.2, 2.3, -0.1) | (4e11, 2e12, 5e12) |
> > > |Suggested zero initialization| (4.1, -0.3, -0.7) | (6e11, 3e13, 1e11) |
> > >
> > >
> > > We thank you again for your response and for your recommendation of acceptance for our paper!

---

> > > > ### Comment · Reviewer_8ru8 · 2024-08-10
> > > >
> > > > Thank you for the speedy response and running this extra experiment. I didn’t actually mean for you to run it, I just wanted to make you aware of the possible initialisation. But nevertheless, interesting to see that this also doesn’t help.
> > > >
> > > > Hopefully the other reviewers engage soon and we can discuss things further.

---

### Official Review · Reviewer_GKpt · 2024-07-01

**Soundness:** 3
**Presentation:** 3
**Contribution:** 2
**Rating:** 5
**Confidence:** 4

**Summary:**

The paper presents a novel analysis of the challenges faced by offline model-based reinforcement learning (RL) when the dynamics model becomes increasingly accurate. The central thesis is that existing methods fail under true, error-free dynamics due to a previously unaddressed issue termed the "edge-of-reach" problem. The authors provide a thorough investigation, theoretical analysis, and empirical evidence to support their claims. They propose Reach-Aware Value Learning (RAVL), a method that directly addresses this problem, demonstrating robust performance across benchmarks.

**Strengths:**

1. The paper is well-organized, with a clear abstract, introduction, and conclusion that effectively summarize the contributions and findings.

2. The authors have provided open-source code and detailed hyperparameter settings, facilitating the reproduction of the experiments.

3. Rational experiments on standard benchmarks and a simple environment validate the "edge-of-reach" hypothesis and demonstrate the effectiveness of RAVL.

**Weaknesses:**

The methodology contribution of this paper is limited. While I assume that the "edge-of-reach" problem is significant to the current offline RL literature, the proposed method by the authors fails to demonstrate obvious superiority compared with the SOTA offline model-free or model-based methods (see Table 2 of this paper). One reason, I guess, is that the D4RL benchmark is too simple to underscore the superiority of addressing the "edge-of-reach" problem. On the other hand, I expect that authors can validate their method on more complex and challenging benchmarks and provide more convincing results. I am willing to reconsider my score after seeing this.

**Questions:**

1. I wonder how existing uncertainty-penalty-based offline model-based RL methods perform if I gradually increase the length of rollouts in the estimated dynamics model along with the improved accuracy of the model (a very intuitive trick).

2. Does the "edge-of-reach" hypothesis exist in many online RL scenarios? For example, researchers usually early stop the trajectory according to a pre-defined maximal length of the trajectory in the Gym Mujoco benchmark.

**Limitations:**

The authors have discussed the limitations of their work properly.

---

> ### Author Rebuttal · Authors · 2024-08-07
>
> We would like to thank the reviewer for their careful feedback on our submission.
>
> We are grateful that they appreciated our “thorough investigation, theoretical analysis, and empirical evidence to support [our] claims” and “robust performance across benchmarks.”
>
> We found their questions particularly insightful (see below).
>
> ---
>
> **W1.1 - Performance on D4RL**
>
> The reviewer highlights that, on the D4RL benchmark, we only match SOTA rather than surpassing it. Below we discuss why this is expected and why we believe this does not affect the key contribution of our paper.
>
> In Section 6.5 we describe why existing methods can sometimes be made to work despite misunderstanding the underlying issues in offline RL. (tl;dr: If there is a lucky correlation between dynamics uncertainty and EOR states, then the dynamics penalties of existing methods can indirectly address the true EOR problem.) For the common D4RL setup (on which many existing methods have been developed) this lucky correlation holds (see Figure 6), meaning existing methods can be made to work well, and we would not necessarily expect RAVL to beat them. (Further, the majority of SOTA scores are close to, or already exceed, expert level $>100$.)
>
> In general, however, this lucky correlation is not guaranteed, and the indirect approach of existing methods means we may expect them to be very brittle to dynamics model changes. Indeed, in Figure 1 of the rebuttal PDF, we show that existing methods completely break down if the dynamics model is changed slightly. RAVL, by contrast, is much more robust due to directly addressing the true underlying EOR problem (see Figure 1 in the rebuttal PDF).
>
> We believe the correction of a significant misunderstanding, and RAVL’s consequent much-improved robustness are highly valuable contributions to the community.
>
> **W1.2 - Performance on more complex and challenging benchmarks**
>
> We thank the reviewer for asking about more complex and challenging benchmarks. We would like to highlight our additional results on the pixel-based V-D4RL benchmark in Appendix F (moved to the Appendix due to space limitations).
>
> Here RAVL represents a new SOTA, giving a performance boost of more than 20% for some environments. These results are particularly notable as the pixel-based setting means the base algorithm (DreamerV2) uses model rollouts in an imagined latent space (rather than the original state-action space as in MBPO). The results therefore give promising evidence that RAVL is able to generalize well to different representation spaces.
>
> We will use the camera-ready extra page to feature these results in the main paper. We hope this answers the reviewer’s concern with a much higher-dimensional and more challenging benchmark.
>
> **Q1 - What happens if we increase the rollout length as dynamics accuracy improves?**
>
> We thank the reviewer for this interesting and insightful question. Tuning the rollout length as suggested is indeed what we tried first.
>
> In the limit of perfect dynamics accuracy this of course works, since rollout length can be increased to full horizon and the procedure becomes online RL. For intermediate dynamics model accuracies, however, we experimented with this extensively but found we were unable to get existing methods to work.
>
> Our intuition for why this is is as follows:
>
> Consider a 2D environment like in Section 4, and imagine we have a dynamics model for which error and uncertainty is significant to the left, and negligible to the right. The model being imperfect means rollouts will need to be truncated. (All existing methods heavily truncate rollouts e.g. $k=1,5$ to avoid compounding errors, even with dynamics uncertainty penalties). However, rollout truncation means we now have EOR states.
>
> The question is: *Is it possible to tune dynamics uncertainty-based methods to work in this setting?*
>
> Uncertainty penalties can successfully penalize the EOR states to the left. However, it is impossible to get uncertainty penalties to target the EOR states to the right (since model uncertainty to the right is negligible).
>
> This illustrates that we can have dynamics models for which it is theoretically impossible to get uncertainty-based methods to work, even allowing for tuning the rollout length. (RAVL, by contrast, would be expected to work in this setting since it directly ensures all EOR are targeted.)
>
> The much stronger reliance on the dynamics model for uncertainty-based methods compared to RAVL (as highlighted by this setting) fits with our empirical observations of existing methods being much more brittle to changes in the dynamics model. See for example our new results (Figure 1 of the rebuttal PDF), in which we find that RAVL is significantly more robust, both when model accuracy is increased and decreased.
>
> We thank the reviewer for asking this. We found it to be a very interesting and insightful question and would be more than happy to discuss further.
>
> **Q2 - Does the "edge-of-reach" hypothesis exist in many online RL scenarios?**
>
> Thank you for another insightful question!
>
> The trajectory cutoff, for example $H=1000$ for MujoCo, can indeed be viewed as a truncated horizon. This means the EOR problem could in theory be present in online RL. We anticipate, however, that $H=1000$ is sufficient for the agent to be able to reach all possible states within this horizon, meaning that the set of EOR states will be empty and the EOR problem will not be present.
>
> As briefly mentioned in the limitations section, however, we are interested in investigating whether truncated horizons could be used in some way in online RL to harness the EOR effect and induce an implicit exploration bias.
>
> ---
>
> We thank the reviewer again for their valuable feedback and insightful questions. We hope we have addressed all of your concerns. Please let us know if you have any remaining questions. If we have been able to address your concerns we humbly ask if you would recommend acceptance.

---

> > ### Comment · Reviewer_GKpt · 2024-08-12
> >
> > Thank you for your response -- I appreciate all the explanations and additional results. I'm raising my score to 5.

---

> > > ### Author Response · Authors · 2024-08-12
> > >
> > > Thank you for your reply, and again for your helpful feedback that has helped us improve our paper. We deeply appreciate you raising your score. Please let us know if you have any remaining concerns and if there is anything we can provide in the discussion period that would enable you to raise your score further.

---

### Official Review · Reviewer_PtsT · 2024-07-10

**Soundness:** 3
**Presentation:** 2
**Contribution:** 3
**Rating:** 7
**Confidence:** 4

**Summary:**

This paper identifies and investigates a previously neglected issue in offline model-based RL called the "edge-of-reach problem", which is due to the truncated rollouts used in model-based RL to mitigate the compounding errors of the dynamics model. The authors proposed Reach-Aware Value Learning (RAVL) to address the edge-of-reach problem using value pessimism. They show RAVL's strong performance on the D4RL and V-D4RL benchmarks.

**Strengths:**

1. The edge-of-reach problem is interesting and previously overlooked in offline model-based RL. This paper is the first to formally investigate the problem.
2. Comprehensive, well-designed experiments to support the claims.

**Weaknesses:**

1. The proposed method doesn't outperform existing baselines according to the reported results on D4RL. Although it's claimed that "RAVL can be easily combined with appropriate dynamics uncertainty penalization", it's not supported by any empirical evidences. Hence, it's unclear how well RAVL work in general when combined with other model-based methods with design choices orthogonal to value pessimism.
2. I found the paper a bit hard to read due to the formatting, e.g. too many long italics phrases.

**Questions:**

1. Why does RAVL not benefit from reduced dynamics errors according to table 1?
2. It would be interesting to show how RAVL's performance change by varying the accuracy of the dynamics model, similar to figure 1.

**Limitations:**

Yes

---

> ### Author Rebuttal · Authors · 2024-08-07
>
> We thank the reviewer for their careful review of our submission.
>
> We are encouraged that the reviewer found the problem to be “interesting,” and the experiments “comprehensive, well-designed”, and are grateful they appreciated that the work is “the first to formally investigate the problem”. We have addressed their questions and comments below.
>
> ---
>
> **W1 - Combining RAVL with dynamics uncertainty penalization**
>
> The reviewer asked for support of the claim that "RAVL can be easily combined with appropriate dynamics uncertainty penalization.”
>
> In terms of implementation: this is extremely simple since dynamics penalization is typically a change to the dynamics model’s reward, while RAVL’s value pessimism is a change to the agent’s value function update.
>
> In terms of empirical evidence: we refer the reviewer to the V-D4RL experiments. As stated in Appendix C.3, for the RAVL results we add RAVL’s value pessimism in addition to the dynamics pessimism already used in the base procedure. As such, the V-D4RL results are an example of combining RAVL’s value pessimism with dynamics uncertainty penalization, exactly as requested. This provides evidence that RAVL’s value pessimism is effective even when the model is less accurate and dynamics uncertainty is necessary. We will make the details in Appendix C.3 more prominent in the camera-ready submission.
>
> **W2 - Formatting e.g. italics affecting readability**
>
> Having looked over the paper again we completely agree with this. We will make sure to refine our use of italics and bold text etc. ahead of the camera-ready submission. Thank you for bringing this to our attention.
>
> **Q1 - Why does RAVL not benefit from reduced dynamics errors according to Table 1?**
>
> Thank you for this question.
>
> Section 3.4 claims and gives evidence that dynamics model errors are already negligible for the main D4RL benchmark (and that issues are instead caused by the edge-of-reach problem).
>
> The trend referred to (namely that reducing dynamics errors does not affect performance) is therefore exactly what we would expect, and the results in Table 1 can hence be seen as further evidence that (contrary to common understanding) dynamics errors contribute negligibly to the issues seen in offline model-based RL.
>
> **Q2 - Results of RAVL's performance with different dynamics model accuracies**
>
> [Please see the rebuttal PDF.]
>
> We have run additional experiments (4 seeds), and have added results for RAVL to the original Figure 1 as requested (please see the new Figure 1 in the rebuttal PDF). As expected, RAVL’s performance remains strong as dynamics model accuracy increases. Furthermore, RAVL appears to be significantly more robust than dynamics uncertainty-based methods as dynamics model accuracy is reduced.
>
> We sincerely thank you for asking for this. These results significantly strengthen our paper and we are excited to add them to the camera-ready submission.
>
> ---
>
> We thank the reviewer again for their invaluable feedback. We hope we have addressed all of your questions and concerns. Please let us know if there is anything remaining as we would be more than happy to discuss further. If we have been able to address your concerns, we humbly ask if you would consider raising your score.

---

> > ### Comment · Reviewer_PtsT · 2024-08-13
> >
> > Thank you for the response and the additional experiments. I have increased my score.

---

> > > ### Author Response · Authors · 2024-08-14
> > >
> > > Thank you for your reply and for rasing your score. We really appreciate your support of our paper.

---

### Official Review · Reviewer_biRH · 2024-07-16

**Soundness:** 3
**Presentation:** 3
**Contribution:** 3
**Rating:** 7
**Confidence:** 3

**Summary:**

The paper investigates a phenomena in model-based offline RL that most existing algorithms fail when they are given the perfect model of the environment. Through multiple experiments, the paper argues the failure is due to the wrong estimated values at the states in the end of model rollouts. These states are called edge of reach and are used for targets of Q values but are never trained themselves. The paper continues to propose a fix in the form of pessimistic target value estimate using ensemble Q networks.

**Strengths:**

- The paper does an admirable job with empirical investigations of the root cause of the observed phenomenon.  I find the perfect model experiments, the simple environment example, and the patch oracle fix to be a very convincing set of evidences for the paper's claim. The paper sets a good example for this kind of empirical research.
- The introduced fix is practical and simple, and I believe might improve other existing model-based offline RL algorithms as well
- The empirical results obtained by the
- I find the paper well written and free of typos.

**Weaknesses:**

**1)** The paper could benefit from more investigations on the necessity of the _detection_ of an edge-of-reach state. The current proposed value learning only uses the pessimistic Q target when it decides that the state is an edge-of-reach state.

**Q1)** I wonder whether if this distinction is necessary and if we could always use the pessimistic value and get the similar results.

I believe this could be doable because in states with low variance, the pessimistic value will not defer from the normal one anyway. Potentially, this removes a hyperparameter which sets the threshold for the choice of value target. It also reduces the number of ensemble networks needed for the algorithm.

**2)** The paper could better position itself in the literature. The pessimistic value target is a ubiquitous trick in RL to overcome value estimation. See for example TD3 [1] paper for actor critic methods. The issue where models query Q networks in untrained states has also been pointed out in MBRL research. See for example [2].

**3)** It probably is out of the scope of the paper, but I wonder how much of this issue is shared with online MBRL algorithms. They also usually only create rollouts of a limited length. A discussion on this matter will be welcome.

**4)** It could be a personal matter, but the paper has too many future referencing. For example lines 160, 220, 231, 253, 260, 261. I find these confusing as a reader. Jumping to read the future section while reading is not ideal or even possible. Also when the future section is being read, the connection made in the past sections to the current one is probably forgotten. I recommend reducing these references and only make such connections when necessary. A general comment that more discussions about a topic will appear in the future could be enough for the reader without asking them to jump.


[1] Fujimoto, Scott, Herke Hoof, and David Meger. "Addressing function approximation error in actor-critic methods." International conference on machine learning. PMLR, 2018.
[2] Jafferjee, Taher, et al. "Hallucinating value: A pitfall of dyna-style planning with imperfect environment models." arXiv preprint arXiv:2006.04363 (2020).

**Questions:**

See Q1, and weakness 3 above.

**Limitations:**

I find the limitation discussions sufficient.

---

> ### Author Rebuttal · Authors · 2024-08-07
>
> We would like to thank the reviewer for their valuable feedback on our submission.
>
> We are particularly encouraged by their appreciation of the care we took with the empirical investigations, in particular with the comments that we provide “a very convincing set of evidences for the paper's claim,” and “set a good example for this kind of empirical research”.
>
> ---
>
> **W1/Q1 - “necessity of the *detection* of an edge-of-reach state”**
>
> The suggestion of removing the explicit “detection” step is in fact exactly how RAVL is implemented (see Equation 2). We just chose to describe it as (A) detecting and (B) penalizing edge-of-reach states to provide intuition of what RAVL is effectively doing. Thank you for this question. We will make it clearer in the paper that this two-phase description is just for intuition purposes.
>
> **W2 - Position in literature and additional references [1] and [2]**
>
> Thank you for highlighting this. We have referenced [1] briefly in the related work section already, but we will use the camera-ready extra page allowance to more thoroughly discuss value pessimism in online RL, including the references [1] and [2] you suggested.
>
> **W3 - Effect of edge-of-reach states in *online* MBRL**
>
> Thank you for this question. Yes, we are also interested in extending this work to online MBRL. One key distinction here is the pool of initial rollout states. In *offline* MBRL this is fixed over the whole of training ($s_0 \sim \mathcal{D}_\text{offline}$), while in *online* MBRL this changes as the agent collects additional trajectories online. The fixed initial state distribution is an important aspect of edge-of-reach states (see Definition 1). There may, however, be a somewhat analogous notion of edge-of-reach states in online MBRL. As briefly mentioned in the limitations section, we would be interested to investigate whether this could be used (or perhaps already plays a role) as an implicit exploration bias in online MBRL, and believe this could be an exciting direction for future work.
>
> **W4 - Future referencing**
>
> Thank you for bringing this to our attention. We agree that the current future referencing may be confusing. Your pointer of instead notifying the reader that “more detail *will* appear in the future” rather than asking them to jump, is extremely helpful and we will definitely make this change. We plan to also use the camera-ready extra page allowance to remove future referencing where possible.
>
> ---
>
> We thank the reviewer again for their valuble feedback. The pointers (in particular about style of future referencing) are really helpful. We hope we have addressed all of your questions. Please let us know if any further discussion would be useful as we would be more than happy to answer additional questions.

---

> > ### Comment · Reviewer_biRH · 2024-08-13
> > **Rebuttal Acknowledgement**
> >
> > I thank the authors for the rebuttal. I have read all the comments and I am maintaining my score.

---

> > > ### Author Response · Authors · 2024-08-14
> > >
> > > Thank you for your reply. We are deeply grateful for your continued support of our paper.

---

### Author Rebuttal · Authors · 2024-08-07

We would like to thank the reviewers. It is clear they have each taken time and care, and their feedback has greatly helped us improve and refine our work.

We are gratified by the reviewers' overall positive assessment of our work, including comments that the paper “should be accepted,” "will be an important contribution to the community" (Reviewer 8ru8), and the appreciation that “this paper is the first to formally investigate the problem” (Reviewer PtsT).

We are also pleased that the reviewers appreciated RAVL’s “robust performance across benchmarks” (Reviewer GKpt), found the set of experiments to be “a very convincing set of evidences for the paper’s claim,” and found the paper to “set a good example for this kind of empirical research” (Reviewer biRH).

While reviewers commented that the paper was “well-organized” (Reviewer GKpt), and “well written” (Reviewer biRH), the key shared feedback seemed to be that editing the formatting (punctuation, italics, forward referencing, etc.) would make the paper easier to read. We thank the reviewers for this feedback and will be sure to make these changes ahead of the camera-ready submission, especially with the extra page allowance.

**New results showing RAVL’s robustness with changing dynamics accuracy**

[Please see the Rebuttal PDF.]

We are excited to present additional results (as requested by Reviewer PtsT) where extend Figure 1 of the paper to include RAVL. The original version of Figure 1 showed how dynamics uncertainty-based methods fail as dynamics accuracy is increased (in red). Please see Figure 1 of the Rebuttal PDF where we have now added RAVL (in purple).

The new results show that, while existing methods fail, RAVL maintains performance as dynamics accuracy is increased. Additionally, in the other direction, we also find that RAVL is more robust to decreasing dynamics model accuracy.

This observation of RAVL’s much stronger robustness to dynamics model changes fits with our understanding of the edge-of-reach problem, since existing methods address the edge-of-reach problem indirectly via relying on the uncertainty of the dynamics model, while RAVL directly addresses the problem without relying on the dynamics model’s uncertainty. We are excited to share these new results as they further highlight the strength of our approach.

---

### Comment · Area_Chair_nu7J · 2024-08-11
**Please respond to author response if you haven't**

Dear Reviewers,

Before the author-reviewer discussion period ends, please make sure that you have read the author responses, acknowledge that your review reflects them, and engage with the authors if necessary.

Thank you!

---

### Decision · Program_Chairs · 2024-09-25

**Decision:**

Accept (poster)

**Comment:**

This paper explores the so-called "edge-of-reach" problem in offline model-based RL, which arises when using short, truncated rollouts to generate synthetic data. Contrary to prevailing theories, the paper finds that improving the learned dynamics model does not necessarily enhance performance; in fact, using an error-free dynamics model can lead to the complete failure of existing methods. This failure is due to the existence of "edge-of-reach" states, where value estimates become unreliable because these states are only encountered at the end of rollouts and not during intermediate steps. To address this issue, the paper proposes Reach-Aware Value Learning (RAVL), a method that directly targets the edge-of-reach problem, thereby preventing fatal value overestimation resulting in the collapse of performance. RAVL demonstrates robust performance even as dynamics models improve, offering a more future-proof approach to offline RL.

The reviews are overall positive, and thus recommend to accept the paper. Authors are encouraged to incorporate additional results provided during the dicussion phase.